# Leiomyogenic Tumor of the Spine: A Systematic Review

**DOI:** 10.3390/cancers16040748

**Published:** 2024-02-10

**Authors:** Abdurrahman F. Kharbat, Kishore Balasubramanian, Kiran Sankarappan, Ryan D. Morgan, Khawaja M. Hassan, Paolo Palmisciano, Panayiotis E. Pelargos, Michael Chukwu, Othman Bin Alamer, Ali S. Haider, Tarek Y. El Ahmadieh, John F. Burke

**Affiliations:** 1Department of Neurosurgery, University of Oklahoma Health Sciences Center, Oklahoma City, OK 73104, USA; pelargos.panayiotis@gmail.com (P.E.P.); john-burke@ouhsc.edu (J.F.B.); 2Division of Neurosurgery, Texas A&M University College of Medicine, Bryan, TX 77807, USA; kishoreb521@gmail.com (K.B.); kiranvsankar@tamu.edu (K.S.); 3Division of Neurosurgery, Texas Tech University Health Sciences Center, Lubbock, TX 79430, USA; ryan.d.morgan@ttuhsc.edu; 4Department of Neurosurgery, King Edward Medical University, Lahore 54000, Pakistan; muthammirkhawaja@gmail.com; 5Department of Neurosurgery, University of California Davis, Davis, CA 95616, USA; paolopalmisciano94@gmail.com; 6Department of Surgery, New York Presbyterian Hospital/Weill Cornell, New York, NY 10065, USA; chukwumichaeleze@gmail.com; 7Department of Neurosurgery, University of Pittsburg Medical Center, Pittsburg, PA 15219, USA; binalameroa@upmc.edu; 8Department of Neurosurgery, The University of Texas MD Anderson Cancer Center, Houston, TX 77030, USA; aralam09@gmail.com; 9Department of Neurosurgery, Loma Linda University, Loma Linda, CA 92354, USA; telahmadieh@gmail.com

**Keywords:** spine leiomyoma, spine leiomyosarcoma, leiomyogenic tumor, en bloc resection leiomyoma, leiomyoma spine resection, spine leiomyoma immunohistochemistry

## Abstract

**Simple Summary:**

Due to limited published data and a lack of formal guidelines, the management of leiomyogenic spine tumors (LTS) is challenging. Here, we report the clinical characteristics of patients with LTSs, analyze treatment modalities and outcomes, and highlight prognostic factors associated with morbidity and survival. Additionally, we endeavor to delineate the efficacy of en bloc resection versus other surgical techniques and understand the impact of surgical techniques on surgical outcomes. In this study, we performed a systematic review of the literature to encapsulate the clinical characteristics of patients afflicted with LTSs, analyze treatment modalities and outcomes, and highlight prognostic factors that inform clinicians on morbidity and survival.

**Abstract:**

The study cohort consisted of 83 patients with a mean age of 49.55 (SD 13.72) with a female preponderance (60 patients). Here, 32.14% of patients had primary LTS; the remaining were metastases. Clinical presentation included nonspecific back pain (57.83%), weakness (21.69%) and radicular pain (18.07%). History of uterine neoplasia was found in 33.73% of patients. LTS preferentially affected the thoracic spine (51.81%), followed by the lumbar (21.67%) spine. MRI alone was the most common imaging modality (33.33%); in other cases, it was used with CT (22.92%) or X-ray (16.67%); 19.23% of patients had Resection/Fixation, 15.38% had Total en bloc spondylectomy, and 10.26% had Corpectomy. A minority of patients had laminectomy and decompression. Among those with resection, 45.83% had a gross total resection, 29.17% had a subtotal resection, and 16.67% had a near total resection. Immunohistochemistry demonstrated positivity for actin (43.37%), desmin (31.33%), and Ki67 (25.30). At a follow-up of 19.3 months, 61.97% of patients were alive; 26.25% of 80 patients received no additional treatment, 23.75% received combination radiotherapy and chemotherapy, only chemotherapy was given to 20%, and radiotherapy was given to 17.5%. Few (2.5%) had further resection. For an average of 12.50 months, 42.31% had no symptoms, while others had residual (19.23%), other metastasis (15.38%), and pain (7.69%). On follow-up of 29 patients, most (68.97%) had resolved symptoms; 61.97% of the 71 patients followed were alive.

## 1. Introduction

Leiomyomas are a heterogeneous group of mesenchymal tumors. They are composed of smooth muscle of various etiologies, most commonly uterine, and are generally benign [1,2,3,4]. Leiomyomas are generally asymptomatic and have a prevalence rate of 70% in white women and more than 80% in women of African descent [3]. In this study, we will be discussing both primary and metastatic leiomyogenic tumors. While most leiomyogenic metastases occur in the lung, they have also been reported to metastasize to the cervical, thoracic, lumbar, and sacral spine [1,2,4,5,6,7]. Leiomyogenic tumors of the spine (LTS) frequently manifest with new neurological deficits and are most often detected via T1 and T2 weighted MRI. The hallmark radiologic feature of an LTS is a well-circumscribed mass, often located on the posterior vertebral body, with low signal intensity and substantial homogenous enhancement. Bony destruction, neural foramen invasion, and/or canal encroachment can also be seen [7]. Pathological specimens of LTS are characterized by a diffuse infiltration of uniform spindle cells with fascicular arrangement, low mitotic index (Ki 67 index often <1%), and immunohistochemical staining demonstrating positivity for smooth muscle actin and desmin and negativity for neurofilament, S100, and epithelial membrane antigen [8].

Depending on the location of the lesion, various neurological deficits may arise, ranging from radiculopathies to spinal cord compression [9,10]. Confounding variables, such as immune status, may also affect the tumor burden of patients. Immunocompromised individuals are predisposed to the development of multifocal tumors due to impaired immune surveillance. As such, they typically present with a wide range of concomitant neurological and physical deficits depending on tumor location [11]. Current therapeutic strategies are directed at relieving tumor burden through en bloc or piecemeal surgical resection and radiotherapy, followed by hormonal treatment and radiological surveillance [2,10,12,13]. Ultimately, treatment goals are established on a case-by-case basis due to the high rate of recurrence.

In this study, we performed a systematic review of the literature to encapsulate the clinical characteristics of patients afflicted with LTSs, analyze treatment modalities and outcomes, and highlight prognostic factors that inform clinicians on morbidity and survival.

## 2. Materials and Methods

### 2.1. Literature Search

A systematic review was conducted on the Preferred Reporting Items for Systematic Reviews and Meta-Analyses (PRISMA) guidelines [14]. PubMed, Scopus, Web of Science, and Cochrane were searched from database inception to 17 September 2023 operating the Boolean full-text search [[leiomyoma OR leiomyogenic] AND [spine OR cervical OR thoracic OR lumbar OR sacral]]. Studies were exported to Mendeley, and duplicates were deleted.

### 2.2. Study Selection

The study established specific criteria for inclusion and exclusion. Articles were considered eligible for inclusion if they met the following conditions:Included at least one patient diagnosed with a leiomyogenic tumor affecting the spine as a primary tumor or originating from a primary non-spinal leiomyoma, based on radiological, clinical, or pathological diagnosis;Reported data on clinical aspects, treatment, and outcomes;Were written in the English language.

On the other hand, studies were excluded if they:Were reviews, autopsy reports, or focused on animal studies;Involved patients with spinal LMs originating from primary CNS tumors;Failed to differentiate data of patients with spine LMs from those with only intra-cranial LMs;Lacked sufficient data regarding treatment outcomes.

All titles and abstracts were screened, and full texts of articles meeting the inclusion criteria were assessed by two independent reviewers, A.F.K. and K.B. Disagreements were resolved through the involvement of a third reviewer, P.P. Eligible papers were included, and references were also screened to identify any additional relevant studies.

### 2.3. Data Extraction

Two reviewers (A.F.K. and K.B.) extracted data from each article, which were then confirmed independently by two additional reviewers (P.P. and O.B.A.). The authors did not report missing data.

Extracted data included author, sample size, age, gender, primary presenting symptoms, duration of symptoms, comorbidities, history of uterine neoplasia, imaging modality used at diagnosis, tumor location, surgery type, surgical complications, resection amount, biopsy results, tumor origin (primary vs. metastasis), cytology, immunohistochemistry (IHC) sensitivity, postoperative imaging, adjuvant therapies, clinical assessment at initial follow-up, additional interventions, last follow-up, symptom resolution, and survival status at last follow-up. Clinical and radiological responses were assessed at the most recent follow-up. Clinical responses were determined by comparing post-treatment functional quality to the pre-treatment status, as reported by the authors. This evaluation included categories such as resolution of baseline symptoms, improvement of baseline symptoms, no change in baseline symptoms (stable), and worsening of baseline symptoms.

### 2.4. Data Synthesis and Quality Assessment

The main variables of interest included the clinical characteristics, management strategies, and treatment outcomes of patients with LTS. Two independent authors, A.F.K. and K.B., evaluated the level of evidence for each study using the 2011 Oxford Centre For Evidence-Based Medicine guidelines [15]. Additionally, the risk of bias was assessed using the Joanna Briggs Institute checklists for case reports and case series [16]. The possibility of conducting a meta-analysis was ruled out due to all included studies having levels IV-V of evidence, and hazard ratios could not be derived.

## 3. Results

### 3.1. Study Selection and Overview

The search strategy yielded 3954 studies (PubMed: 1977. Web of Science: 663. Scopus: 1676. Cochrane: 1), of which 46 were included in the qualitative synthesis (Figure 1). In total, 6 were case series (including 43 patients), and 40 were case reports, with IV and V levels of evidence, respectively (Appendix A) [17,18,19,20,21,22,23,24,25,26,27,28,29,30,31,32,33,34,35,36,37,38,39,40,41,42,43,44,45,46,47,48,49,50,51,52,53,54,55,56,57,58,59,60,61,62].

### 3.2. Demographics and Primary Tumors’ Characteristics

A total of 83 patients diagnosed with LTS were analyzed (Table 1). The mean age at diagnosis was 49.55 years (SD 13.72), and 72.29% (n = 60) were female. Medical comorbidities were reported in 6.02% (n = 5) of the patient population, and two patients (2.41%) had a history of HIV. Additionally, 33.73% (n = 28) of the patients had a prior history of uterine neoplasia. The thoracic region was the most common tumor location, occurring at 51.81% (n = 43). This was followed by the lumbar spine, with 21.67% (n = 18) of tumors in this region. LTS was associated with metastatic disease in 67.86% (n = 38) of cases. Back pain was the most common presenting symptom, with 57.83% (n = 48) of patients endorsing back pain of some degree. Other presenting symptoms included lower and upper extremity pain, weakness, and other neurologic symptoms.

### 3.3. Clinical and Diagnostic Features of Spine LTS

The imaging modalities used in diagnosing LTS varied by patient, though the most common were MRI alone (33.33%, n = 16) and a combination of CT and MRI (22.92%, n = 11). Despite the wide variety of imaging modalities, 91.67% (n = 44) of patients subjected to imaging studies received an MRI, highlighting the importance of MRIs in the radiological workup (Table 1). In total, 32% (n = 18) of LTSs were primary tumors, whereas 67.86% (n = 38) were metastases. The most common location of the primary tumor site was the uterus, present in 33.73% (n = 28). Furthermore, upon biopsy, 48 were identified as leiomyosarcomas, and 13 were identified as benign leiomyomas. Post-biopsy, actin was positive in 43.37% (n = 36), desmin was positive in 31.33% (n = 26), and Ki67 was present in 25.30% (n = 21). S100, however, was negative in 24.10% (n = 20) (Table 2).

### 3.4. Management Strategies

Surgery was the primary treatment modality for 78 patients (Table 3). Resection and Fixation was the most commonly surgical approach in 19.23% (n = 15) of patients. This was followed by total en bloc spondylectomy in 15.38% (n = 12) of patients. Surgical complications were rare, only occurring in 4.82% (n = 4) of patients. Complications included thrombocytopenia, injury to the vertebral artery requiring repair, a pleural defect, and a deep vein thrombosis in the left lower extremity [19,49,50]. Among 24 patients who had a resection of the LTS, 11 had a gross total resection, and 4 had a near gross total resection. Rarely was surgery the only treatment methodology utilized, and adjuvant therapies played an important role in the management. The most common adjuvant therapy included radiation and/or chemotherapy (Table 4). One patient was treated with tamoxifen, another was treated with letrozole for one month, and another was treated with a combination of leuprorelin and anastrozole [29,42]. 

### 3.5. Treatment Outcomes and Survival

Long-term survival data were available for 71 of the 83 patients. Of these 71, there was a mortality of 38.03% (n = 27) (Table 4). The average initial and latest follow-up times were 12.5 (SD 9.83) and 19.29 (SD 24.69) months, respectively. Four patients returned with acute worsening neurologic symptoms, including leg weakness and paralysis. Repeat operations and revision surgery successfully led to a return to baseline in all patients [22,28,35]. One patient had disease progression, which required a hysterectomy and bilateral salpingo-oophorectomy, and a different patient required revision operations and repeated chemotherapy plus radiation [20,22,26]. In total, 14 patients required follow-up interventions, nine of whom received surgical intervention [20,26,33,41,49,50,61]. Symptomatic response to treatment was recorded in 29 of the included patients. Of these 29 patients, 20 (68.97%) had complete symptomatic resolution after surgery. Additionally, the data demonstrate that en bloc resection did not provide any statistically significant advantage when compared to non en bloc methods in terms of overall survival (*p* value: 0.055) [Appendix A].

## 4. Discussion

Due to the paucity of data on these relatively rare tumors, leiomyogenic tumors of the spine are challenging to diagnose and treat. Although the pathogenesis is poorly understood, some hypotheses have been postulated. These theories surround associations with infectious agents (Human immunodeficiency virus, Ebstein–Barr virus, and immunosuppression) and angiogenic elements [63,64,65]. This study aims to build on the previous knowledge on leiomyogenic tumors’ clinical presentation, diagnostic features and prognosis, and integrate findings from newer studies. In addition, we explored the different iterations and complexities in management, potentially offering novel strategies for neurosurgeons and oncologists to deal with this rare disease.

To begin with, there was a substantial specificity in the occurrence of this tumor, with the median age being 49 years and a significantly higher incidence in women, especially premenopausal women. These results strengthen and build upon previous studies in spinal leiomyoma, which also report similar trends [66,67,68]. Furthermore, similar patterns were also recognized in women with uterine leiomyomas [69]. One plausible explanation for such a high occurrence in this population could be that the metastases from previous uterine leiomyomas were diagnosed as spinal leiomyoma. Most female patients with primary LTS also reported a previous history of uterine fibroids. These findings raised essential points of discussion regarding the relationship between previous uterine fibroids and primary spinal leiomyoma [70]. Given this relationship, the investigation of the patients with a positive history should be expanded to consider the presence of leiomyomas. Furthermore, this opens an avenue for further research into a possible strong association between uterine leiomyomas and primary LTS in females.

Although the prevalence of medical comorbidities was relatively low, a striking 4.4% of patients had a history of HIV, warranting further investigation between LTS and immunosuppression, as there are hypotheses linking viral etiologies such as HIV and EBV.

The most common symptoms reported were back pain (approximately 58% of cases), spinal cord compression, urinary symptoms, spasms, and neck pain, amongst others. However, these symptoms are usually nonspecific and can present in many other spinal disorders. Thus, a proper history and examination are needed, in addition to consideration of clinical symptoms, to suspect a diagnosis of LTS. The time course of symptoms can also help delineate between differentials as neoplasm-related symptoms evolve over a period of weeks to months. One reason for the high frequency of neurologic symptoms can be attributed to the frequent involvement of the thoracic spine (51.81%) and the lumbar spine (Table 1). Similar locations were also reported in previous studies; however, there seems to be no particular reason for the heightened frequency [70]. Cauda equina syndrome can also be an emergent presentation of these tumors.

The use of an appropriate imaging modality is critical in diagnosis. In our study, MRI with or without a CT scan was the most used radiological technique. Various studies emphasize the importance of periodic surveillance via MRI to monitor progression or recurrence. Lesions on MRI typically show a low signal intensity on T1W/T2W sequences and show contrast enhancement following contrast administration. CT scan alone was rarely used, as reflected in 6.25% of cases. A PET scan is also recommended to differentiate large benign uterine fibroids from leiomyosarcomas. Due to the lack of pathognomonic radiologic features, it may be challenging to make a definitive diagnosis. Hence a multi-modal investigative approach is needed [71]. Using a well-rounded approach comprised of MRI, CT, and PET seems to be the most beneficial as it can help investigate the extent and site of the primary tumor, in addition to proper diagnoses, catering to individual patients.

Another method used to confirm the diagnosis of spinal lesions was histological examination. This helps distinguish leiomyomas from some other common spinal cord tumors. Specific histological markers were also employed to investigate the lesion. Among these, actin, desmin Ki-67, estrogen/progesterone, and caldesmin showed the highest positive rates (43%, 33%, 25%, 12% and 10%, respectively), while others like s100 and CD34 had the highest negative rates [72]. These findings will help future studies and hospital settings develop a better system for diagnosing spinal leiomyoma.

Based on the limited number of studies, the most effective treatment option for alleviating symptoms and improving prognosis remains surgical resection, particularly total en bloc spondylectomy. This approach has also shown promising results in previous studies [6,8]. Alternative treatment options include radiation therapy and embolization. However, these tumors often show resistance to conventional radiotherapy [73]. Despite this, it was commonly used as an adjuvant to surgery with or without chemotherapy. In our study, around 80% of the patients underwent surgery, with the most frequent procedure being laminectomy. This procedure was preferred in cases where the tumor compresses the spinal cord or nerve roots. Resection of the tumor alone is the most widely used approach. Removing the involved vertebral body achieved tumor control in spinal metastatic cancer involving the thoracic and lumbar vertebrae. However, in our analysis we did not see an advantage in pursuing a total spondylectomy. Given the surgical complexity of the procedure, and the inherent risk of further tumor seeding, the risks of en bloc resection far outweigh the potential benefits [74].

A multimodality approach of surgery followed by chemotherapy and stereotactic radiosurgery was often performed. It was based on various factors such as comorbid conditions, age, phase of illness, tumor site, financial constraints, caregiver status, and individual care goals. The mortality rate was 38.03%, which could be attributed to recurrences post-surgery and surgical complications such as thrombocytopenia, vertebral artery injury requiring repair, pleural defect, and deep vein thrombosis.

Thrombocytopenia and deep vein thrombosis highlight the importance of perioperative management, such as thromboprophylaxis, to minimize the risk of clot formation. While vertebral artery injuries requiring repair are a rare complication, they highlight the need for meticulous surgical technique, especially when working near major vascular structures. A pleural defect highlights potential complications beyond the immediate surgical site, emphasizing the need for comprehensive postoperative care and monitoring.

Tumors showing hormone receptor positivity could be treated with hormonal therapy, including tamoxifen, Letrozole, Leuprolide, and Anastrozole [75]. Advanced radiotherapy techniques like Cyberknife and gamma knife can be used to meet the unique needs of the patients [76]. Timely intervention is essential as it can significantly reduce mortality and neurological impairments.

In our study, only three patients received hormonal therapy, which caused tumor regression and reduced the incidence of tumor recurrence.

The median follow-up time reported was 19 months; however, most studies did not report sufficient follow-up data. Most of the patients studied during follow-up were found to be disease-free, reporting no symptoms.

This study has one of the largest sample sizes, despite the rare occurrence of spinal leiomyomas, and was the only study to introduce and discuss the use of immunohistochemical markers in diagnosing spinal leiomyomas.

Our conclusions can be limited as the performed analysis is retrospective, so there might be a chance of selection bias as most are not population-based, and most cases were seen in tertiary centers. Additionally, despite have a large sample size, the data reported in each of the studies were very heterogenous. This made data analysis and synthesis, and the establishment of broad trends, difficult. Due to the lack of randomized control trials, there are no gold standard protocols in managing leiomyogenic tumors of the spine; heterogeneous treatment modalities followed in cases also remain a limitation. Moreover, the full texts were not retrieved for eight articles, which may have aided in proper analysis.

## 5. Conclusions

A multidisciplinary and patient-centered approach is critical in the diagnosis and management of patients with LTS. Integrating medical oncology, surgery, radiation oncology, physical medicine, and rehabilitation can aid in the speedy recovery of patients and minimize neurological complications. Acute and sub-acute rehabilitation programs play a significant role in improving the quality of life and total functional independence of patients after surgery.

Important nuances such as standardized enrollment in clinical trials, considering the patient-reported quality of life, and functional independence, can improve the prognosis of this underserved group of patients. Although LTS are uncommon, they should be regarded as a significant differential diagnosis of spinal cord tumors, especially in females with a history of uterine fibroids. Timely intervention can lead to greater clinical response and a faster recovery of function. As such, neurosurgeons should have a high index of suspicion for this rare tumor. It becomes essential to apply the knowledge gained in devising advanced treatment strategies, as this will benefit the future of therapy. Further investigation and research should be geared towards characterizing the potential relationship between uterine leiomyomas and LTS. While this may not directly impact secondary prevention/direct therapeutics, it will have a major influence on primary prevention and surveillance in high-risk patients.

## Figures and Tables

**Figure 1 cancers-16-00748-f001:**
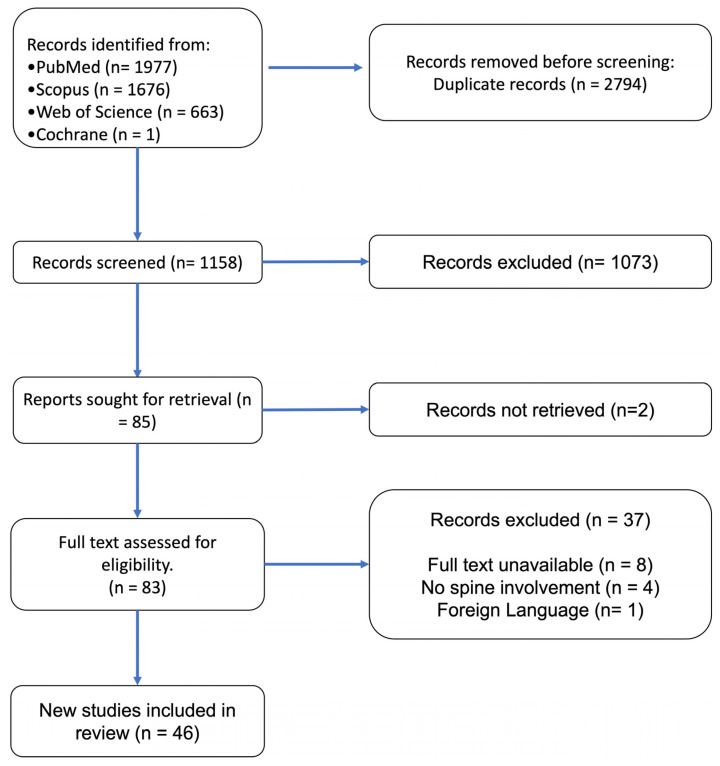
Preferred Reporting Items for Systematic Reviews and Meta-Analyses (PRISMA) 2020 flow diagram.

**Table 1 cancers-16-00748-t001:** Patient demographics and clinical presentation.

Presenting Characteristics (n = 83)	Value [Percentage among Available Data]
Demographics	
Age (y), mean ± SD	49.55 ± 13.72
Sex (n = 83)	Male: 23 [27.71%] Female: 60 [72.29%]
Comorbidities	5 [6.02%]
HIV	2 [2.41%]
Trauma	1 [1.20%]
Past history of uterine neoplasia	28 [33.73%]
Most reported symptoms	
Back pain	48 [57.83%]
Leg Pain	1 [1.20%]
Lower-extremity paralysis	8 [9.64%]
Lower-extremity weakness	18 [21.69%]
Urinary symptoms	4 [4.82%]
Radicular pain	15 [18.07%]
Neck pain	4 [4.82%]
Spasms	1 [1.20%]
Upper-extremity pain	4 [4.82%]
Paraparesis	4 [4.82%]
Paraplegia	2 [2.41%]
Claudication	2 [2.41%]
Hip pain	5 [6.02%]
Other symptoms	2 [2.41%]
Tumor Location (n = 83)	
Thoracic region	43 [51.81%]
Cervical region	6 [7.23%]
Lumbar region	18 [21.67%]
Sacral region	3 [3.61%]
Cervicothoracic	2 [2.41%]
Thoracolumbar	5 [6.02%]
Lumbosacral	4 [4.82%]
Unreported data	2 [2.41%]
Imaging Modalities (n = 48)	
CT alone	3 [6.25%]
MRI alone	16 [33.33%]
CT + MRI	11 [22.92%]
CT + MRI + X-ray	8 [16.67%]
CT + MRI + PET	5 [10.42%]
MRI + X-ray	1 [2.08%]
MRI + PET	2 [4.17%]
MRI + X-ray + PET	1 [2.08%]
CT + X-ray + Myelography	1 [2.08%]
Tumor type (n = 56)	
Primary tumor	18 [32.14%]
Metastatic disease	38 [67.86%]
Biopsy Results (n = 61)	
Leiomyosarcoma	48 [78.69%]
Benign Leiomyoma	13 [21.31%]

**Table 2 cancers-16-00748-t002:** Sensitivities for reported immunohistochemistry markers.

Histological Markers (n = 83)	Value [Percentage among Available Data]		
	Positive	Negative	Not Measured
Actin	36 [43.37%]	1 [1.20%]	46 [55.42%]
Desmin	26 [31.33%]	3 [3.61%]	53 [63.64%]
Caldesmin	7 [10.23%]	3 [3.61%]	73 [87.95%]
CD10	1 [1.20%]	2 [2.41%]	80 [96.39%]
Estrogen/Progesterone	10 [12.05%]	3 [3.61%]	70 [84.34%]
Ki67	21 [25.30%]	2 [2.41%]	60 [72.29%]
Melanoma-Black 45	0 [0.00%]	2 [2.41%]	81 [97.59%]
Collegen IV	1 [1.20%]	0 [0.00%]	82 [98.80%]
p16	1 [1.20%]	0 [0.00%]	82 [98.80%]
p53	1 [1.20%]	0 [0.00%]	82 [98.80%]
s100	1 [1.20%]	20 [24.10%]	62 [74.70%]
SOX10	0 [0.00%]	2 [2.41%]	81 [97.59%]
CD34	2 [2.41%]	6 [7.23%]	75 [90.36%]

**Table 3 cancers-16-00748-t003:** Surgical management strategies and outcomes.

Treatment Data	Value [Percentage among Available Data]
Primary Surgery Type (n = 78)	
Resection + Fixation	15 [19.23%]
Total en bloc spondylectomy	12 [15.38%]
Corpectomy	8 [10.26%]
Laminectomy + Resection	7 [8.97%]
Laminectomy + Fixation	3 [3.85%]
Laminectomy + Decompression	1 [1.28%]
Laminectomy + Fixation + Resection	7 [8.97%]
Resection only	3 [3.85%]
Decompression + Resection	3 [3.85%]
Decompression + Fixation + Resection	3 [3.85%]
Other	16 [20.51%]
Resection amount (n = 24)	
Gross Total Resection	11 [45.83%]
Near-total Resection	4 [16.67%]
Subtotal Resection	7 [29.17%]
Partial Resection	1 [4.17%]
Marginal en bloc Resection	1 [4.17%]
Complications (n = 83)	
No complications reported	79 [95.18%]
Complications reported	4 [4.82%]

**Table 4 cancers-16-00748-t004:** Clinical Outcomes.

Follow Up Data	Value [Percentage among Available Data]
Follow-up	
Average initial follow-up time (n = 26)	12.50 Months ± 9.83
Average last follow-up (n = 63)	19.29 Months ± 24.69
Post Op. Treatment (n = 80)	
Radiation therapy	14 [17.5%]
Chemotherapy	16 [20.00%]
Resection	2 [2.5%]
Radiotherapy and chemotherapy	19 [23.75%]
Other treatment	8 [10.00%]
No treatment	21 [26.25%]
Follow up symptoms (n = 26)	
Pain	2 [7.69%]
Metastasis of cancer	4 [15.38%]
Disease Progression	2 [7.69%]
Follow up masses found	5 [19.23%]
Other symptoms	2 [7.69%]
No symptoms	11 [42.31%]
Follow-up Intervention	
Surgical	9 [66.67%]
Non-surgical	5 [33.33%]
Symptom Resolution (n = 29)	
Symptoms resolved	20 [68.97%]
Symptoms did not resolve	9 [31.03%]
Status (n = 71)	
Alive	44 [61.97%]
Deceased	27 [38.03%]

## Data Availability

All data supporting the reported results can be found in the texts of, or supporting Appendix A of, studies included in our citations.

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
