# Peer review of "Leiomyogenic Tumor of the Spine: A Systematic Review"

_cancers, 2024, doi:10.3390/cancers16040748_

Round 1

Reviewer 1 Report

Comments and Suggestions for Authors

The authors performed a systemic review of report on patients presenting with leiomyogenic tumors involving the spine. This is a rare clinical occurence as leiomyomata are generally benign and do not spread. The study was conducted in accordance with the PRISMA guidelines. I have made specific comments below:

- was the systematic review protocol registered? If so please provide a registration number.

- Table 1 is too long. Please condense or move to Supplemental. A study-by-study summary would make more sense instead of patient-by-patient, but it will likely still be too long at 46 studies.

- Not much synthesis was done on the available data to determine trends otherwise not obvious from the individual case reports/case series. For example, case reports are generally quite detailed when it comes to patient characteristics and outcomes. Specifically, progression or death might have been reported in a large number of studies with the interval between treatment and progression/death. Such data could be used to generate preliminary Kaplan-Meier estimates on the overall cohort. Similarly, baseline characteristics such as age, sex, as well as treatment characteristics such as GTR/STR, use of postoperative treatment would likely have been available, and could be investigated as potential predictive factors of outcomes. These analyses are taken as exploratory, of course, but will represent synthesis of new data and adds further knowledge than previously have existed. If not feasible to further synthesize data, please add a brief statement in the Discussion.

- Is there information on disease-specific survival? It'll likely be more meaningful than OS in the context of this type of tumors.

Reviewer 2 Report

Comments and Suggestions for Authors

The authors described the review about leiomyosarcoma of the spine. The present manusctipt includes important data for physicians possibly treating spinal tumors. Before formal acceptance, the following points had better be considered. 

1) Title. Is the present manuscript "meta-analysis"? Because leiomyosarcoma of the spine is very rare disease, collected papers were case report and/or case series without any control subjects. Therefore meta-analysis cannot be performed. In fact, the present manuscript lacks any of statistical analyses nor comparison with controls. Please consider changing the title.

2) Is there any difference in outcome between surgical methods? There is possibility that total en bloc spondylectomy might result in better outcome because of the nature of the tumor.
